# Design of Transverse Brachiation Robot and Motion Control System for Locomotion between Ledges at Different Elevations

**DOI:** 10.3390/s22114031

**Published:** 2022-05-26

**Authors:** Chi-Ying Lin, Yong-Jie Tian

**Affiliations:** 1Department of Mechanical Engineering, National Taiwan University of Science and Technology, No. 43, Keelung Rd., Sec. 4, Taipei 106, Taiwan; a7ysjd@gmail.com; 2Center for Cyber-Physical System Innovation, National Taiwan University of Science and Technology, Taipei 106, Taiwan

**Keywords:** transverse ledge brachiation, support constraint, gripper design, locomotion control, gripper command prediction

## Abstract

Bio-inspired transverse brachiation robots mimic the movement of human climbers as they traverse along ledges on a vertical wall. The constraints on the locomotion of these robots differ considerably from those of conventional brachiation robots due primarily to the need for robust hand-eye coordination. This paper describes the development of a motion control strategy for a brachiation robot navigating between wall ledges positioned on a level plane or at different elevations. We based our robot on a four-link arm-body-tail system performing a four-phase movement, including a release phase, body reversal phase, swing-up phase, and grasping phase. We designed a gripper that uses passive wrist joint motion to grasp the ledge during the tail swing. We also developed a dynamic model by which to coordinate the swing-up movement, define the phase switching conditions, and time the grasping action of the grippers. In experiments, the robot proved highly effective in traversing between wall ledges of the same or different elevations.

## 1. Introduction

Transverse ledge-climbing robots are a form of bio-inspired climbing robots [1,2,3,4,5,6,7] mimicking the movement of climbers traversing ledges on a vertical wall [8]. They can be used to perform dangerous or labor-intensive tasks, such as surveillance, inspection, rescue, and maintenance [9,10,11,12]. In [13], researchers developed a ledge-climbing robot, which moves horizontally along ledges by bending and extending the arms in a synchronized series of motions; however, short stroke distance limited the range of movement to routes with small gaps between adjacent ledges and only small differences in elevation. The range of motion can be increased by having the arms swing the lower part of the body like a pendulum motion similar to that of primates. This motion is referred to as brachiation [14,15].

Transverse brachiation refers to a swinging movement between ledges on a wall, during which the body is alternately supported under each forelimb. In situations where handholds are beyond the reach of the robot (i.e., non-continuous ledges), the energy stored during the swing phase can be used to leap toward the target ledge (i.e., transverse ricochetal brachiation) [8]. Note that this requires posture compensation during the flight phase to minimize bounce during landing as well as sophisticated eye-hand coordination. Thus, it appears that swinging the robot arm beyond the target ledge for grasping (i.e., overhand brachiation) without a leaping motion would be best suited to ledge-climbing tasks involving ledges at different elevations.

In the current study, we developed a robot that performs transverse overhand brachiation tasks in navigating routes that include multiple ledges at different elevations. The movements of existing transverse brachiation robots with an anterior orientation perpendicular to the direction of movement can be categorized as follows: (1) Grasped object orientation perpendicular to the direction of movement and parallel to the ground [16,17,18,19,20,21,22]; (2) Grasped object orientation perpendicular to the direction of movement and perpendicular to the ground [23]; (3) Grasped object orientation parallel to the direction of movement and parallel to the ground [1,13,24,25,26].

On the premise that the object being grasped is a round bar, most existing studies regard the gripper and the object being grasped as a single unactuated joint (upper arm joint). This assumption simplifies the overall design of the robot and promotes the swinging motion of the lower limb, thereby facilitating system excitation. Brachiation robots that grasp monkey bars automatically perform the desired swing motion through the free rotation of the gripper at the point of contact with the grasped object (round bars). Brachiation robots that grasp ropes or cables use the flexibility of the grasped object to introduce swing motion. Note, however, that this approach is ill-suited to situations involving horizontal ledges [1,13] due to difficulties in maintaining a consistent hand-hold posture in the repeated locomotion cycles, particularly when dealing with complex or unknown grasping dynamics. In the current study, we sought to mimic the up-swing movement of human climbers traversing a ledge without introducing joint actuation. This was achieved by employing a passive wrist mechanism with an active lower limb segment (tail). 

Figure 1 and Table 1 illustrate the characteristics of our proposed ledge brachiation robot as well as the challenges we faced in its development. It is important to consider that brachiation robots are underactuated, which means that there are fewer actuators (control inputs) than degrees of freedom (DOFs). Most existing brachiation robots designed for horizontal locomotion use the aforementioned assumptions on underactuated joints for the sake of simplicity. Very few studies have applied brachiation to the problem of moving from one ledge at a given elevation to another ledge at a different elevation, despite the fact that such actions are common in most practical situations [22,27]. Brachiation performance depends largely on the complexity of contact dynamics between the gripper and ledge. Any deviation in the posture of the grasping device during the last phase of the locomotion cycle can have a deleterious effect on swing efficiency and compromise the ability to perform consecutive brachiation movements. The ability to overcome complex grasping dynamics requires a suitable mechatronic design. 

The robot in the current study uses a four-link model with two arms, one body, and one tail, in which movements are performed in four phases: (1) release phase; (2) body reversal phase; (3) swing-up phase; (4) grasping phase. We also developed a novel gripper with an upper claw that opens and closes horizontally in order to achieve sufficient clamping force using motors with low torque. To ensure that the gripper grasps the target ledge during take-off and landing, we delay the triggering of the grippers to compensate for the time lag between the grasping command and time that the gripper actually closes. In simulations and experiments, the proposed robot succeeded in overhand ledge brachiation while navigating between ledges at different elevations. The contributions of this work are summarized as follows:

1. We developed a ledge brachiation robot that uses an arm-body-tail configuration for transverse movement under tail swing excitation and a novel gripper for rapid power-efficient clamping. 

2. We developed a dynamic robot model to deal with actuator dynamics, the effects of gear backlash, energy accumulation, and grasp timing specifically for ledge brachiation.

3. We present guidelines for the mechatronic design and control of multi-locomotion ledge brachiation robots.

The remainder of this work is organized as follows. Section 2 introduces the proposed robot design and outlines the constraints imposed on a system that lacks actuated joints for the gripper. Section 3 describes the working principle and hardware of the proposed gripper and robot. Section 4 outlines the dynamic model used in motion control. Section 5 outlines the motion control strategies and phase switching conditions used in each locomotion cycle. Section 6 reports on experiments involving transverse ledge brachiation. Conclusions and future research directions are presented in Section 7.

## 2. Robot Design

### 2.1. Locomotion

The proposed robot was meant to mimic the actions of humans moving in a transverse direction along ledges. We also sought to maximize the use of energy generated during swing excitation and allow a range of locomotion styles in adapting to different climbing environments. The proposed brachiation robot includes one body link, two arm links with active shoulder joints, two grippers with passive wrist joints, and one tail link. Figure 2 illustrates the four phases of robot locomotion: (1) release phase, (2) body reversal phase, (3) swing-up phase, and (4) grasping phase.

(1)Release phase:

As indicated by the light-colored plot, the robot lifts the tail and releases the orange gripper from the current ledge to initiate a swing toward the target ledge (Figure 2). Note that lifting the tail stores potential energy for use in the subsequent swing.

(2)Body reversal phase:

Here, we adjust the angle of the arm joints with the aim of rotating the body roughly 180 degrees, such that the orange gripper is positioned on the target side (i.e., moving from left side to right side in Figure 2) while maintaining a tail-down position. Note that while reversing the body from one side to the other side, it is important to account for the distance to the target and the relative position of both arms. As shown in Figure 3, the free hand at coordinates (*x_e_, y_e_*) follows a circular trajectory of radius *d*_2_ when using a simplified dual-link robot with the two arms and the body acting as one link in conjunction with a movable tail as the second link. We assume that the robot maintains a symmetrical body posture during the swing; i.e., the joint angle between the support arm and body (*θ_hb_*) is equal to the joint angle between the body and reaching arm (*θ_br_*). Swing radius *d*_2_ can be derived using the geometric properties of isosceles trapezoids, as follows:(1)$$d_{2}=l_{b}+2l_{h}\cos\theta_{hb}=l_{b}+2l_{r}\cos\theta_{br}$$
where *l_b_* indicates the length of the body, and *l_h_* and *l_r,_* respectively, indicate the lengths of the support arm and reaching arm. Based on the coordinates of the support hand and target (*x_h_, y_h_*) and (*x_d_, y_d_*), reach distance *d*_1_ can be derived as follows:(2)$$d_{1}=\sqrt{{\left|x_{h}-x_{d}\right|}^{2}+{\left|y_{h}-y_{d}\right|}^{2}}$$

Note that the elevation of the target ledge is not necessarily the same as that of the support ledge. If we let *d*_2_ = *d*_1_ and substitute (2) into (1), we obtain joint angle commands *θ_hb_* and *θ_br_* suitable for the swing motion, as follows:(3)$$\theta_{hb}=\theta_{br}=\cos^{-1}\left(\frac{d_{1}-l_{b}}{2l_{h}}\right),\left(0\leq \theta_{hb}\leq \frac{\pi }{2}\right)$$

Our objective in this phase is to achieve the desired robot posture (Figure 3) via a path planning process (outlined in Section 5). 

(3)Swing-up phase:

As shown in Figure 2, the free gripper must be moved to a position above the target ledge. In this phase, the robot maintains the posture achieved in the previous phase while swinging the tail link repeatedly until the swing amplitude is sufficient to initiate brachiation. Note that a swing-up phase is commonly used in underactuated mechanical systems, such as brachiation robots [14,15,16,17,18,19,20,21,22,23,24,25,26,27] or acrobots [28,29,30]. Among the numerous control methods proposed for robot swing motion control [31,32,33,34,35], we adopted the energy-based method proposed by Spong [32] to generate the torque by which to drive the actuated link (tail), such that the direction of the torque vector is the same as the direction of the underactuated link (body and arms). This makes it possible to accumulate energy to increase the amplitude of the swing in a manner reminiscent of a child playing on a swing. For the sake of convenience, we also applied this tail swing strategy to the previous phase to facilitate posture adjustment.

(4)Grasping phase:

Our repetitive swing strategy allows the robot to gradually increase the swing amplitude (under resonant excitation) until the grasping gripper passes beyond the target. Our objective in this phase is to activate the gripper exactly when it intersects with the target location.

### 2.2. Design Constraints

In our analysis above, we assumed that transverse brachiation involves 2D planar motion (frontal view). However, the inability to place the robot’s COG (center of gravity) precisely on the plane of motion can lead to non-negligible deviations in body posture (from a side-view perspective). For example, insufficient gripping force can cause the robot body to deviate from the motion plane and thereby throw the robot off course. Even slight fluctuations in the position of the gripper (shaking) could undermine the storage of swing energy or cause the robot to fall to the ground. Furthermore, the gripper must be able to clamp the target quickly as it passes over the target ledge. 

#### Solution

Figure 4 illustrates the concept underlying the design proposed. The photo showing human hands grasping a ledge in Figure 4a suggests that the gripper could comprise an upper claw (mimicking the four fingers and palm) and a lower claw (mimicking the bent thumb) for added stability. Note that the bio-inspired design in Figure 4b imposes the weight of the entire robot on the upper claw. Under these conditions, the actuator used to open/close the upper claw would require considerable power to control the release of the gripper from the support ledge. We sought to overcome this difficulty by adopting a gripper design in which the upper claw opens and closes horizontally, as shown in Figure 4c. Note that this design reduces the likelihood of inadvertently striking the ledge. Furthermore, limiting the degree of freedom to horizontal motion reduces the load imposed by the weight of the robot. In other words, the weight of the robot is supported by structural supports rather than a closing mechanism. By reducing the torque required for upper claw actuation, it is also possible to create a more compact gripper design.

Note that we also had to overcome the rotation of the claw mechanism along the horizontal plane, which could have a profound effect on the grasping success (see Figure 4d). We adopted a passive wrist mechanism to restore horizontal gripper posture in swinging and grasping and motions.

### 2.3. Selection of Robot Design Parameters

A smooth transition from the body reversal phase to the ledge grasping phase depends on the rigidity and alignment of arm joints. The length of the body link could be extended to increase the grasping distance; however, this would impose a corresponding increase in joint loading. Assume that the maximum grasping distance (equal to the length of the robot body and two arms) is fixed. Further, assume that the robot is a three-link system (i.e., disregard the lower limb mechanism), as shown in Figure 5. Finally, assume that the body link and arms possess total length *L* and mass *M*, and the gripper is a point mass model *m_g_*. *l_c©_*, *i* = 1, 2, 3 indicates the distance between link joints and the center of mass of each link. Assuming that the ratio of body link length *l*_2_ to total link length *L* is *a*, then we can derive the other parameters in Figure 5 as follows:(4)$$\begin{array}{l} l_{2}=aL\\ l_{c2}=\frac{l_{2}}{2}\\ m_{2}=aM\\ l_{1}=l_{3}=\frac{1-a}{2}L\\ l_{c1}=l_{c3}=\frac{l_{3}}{2}\\ m_{1}=m_{3}=\frac{1-a}{2}M \end{array}$$
where *a* ranges from 0 to 1. Our static analysis revealed that the left arm joint requires considerable torque *τ* to maintain horizontal alignment of the body and right arm, regardless of left arm angle *θ*_1_. The torque *τ* required at this moment can be represented as follows:(5)$$\tau =l_{c2}m_{2}g+\left(l_{2}+l_{c3}\right)m_{3}g+\left(l_{2}+l_{3}\right)m_{g}g$$
Substituting (4) into (5) gives the following simplified equation.
(6)$$\tau =\left[\frac{1}{8}{\left(1+a\right)}^{2}LM+\frac{1}{2}\left(1+a\right)Lm_{g}\right]g$$
This analysis indicates that the required torque could be reduced by minimizing the length of the body link. To achieve this, we implemented the actuators and most of the transmission components within the arm links, resulting in a body link length *l*_2_ of 67 mm with the body length to total length ratio *a* = 0.396. Using the above dimensions, the mass of the body link (*m*_2_) was 0.141 kg. 

After selecting the value of *l*_2_, we included the tail link in subsequent modeling as a four-link mechanism, as shown in Figure 6. Again, we assumed that the body link, right arm link, and tail link were aligned horizontally. The torque required to achieve this posture must satisfy the static constraint expressed as follows: (7)$$l_{c2}m_{2}g+\left(l_{2}+l_{c3}\right)m_{3}g+\left(l_{2}+l_{3}\right)m_{g}g+\left(l_{c2}+l_{4}\right)m_{4}g\leq 0.6\tau_{\max}$$
where *τ*_max_ = 2.5 N-m is the maximum torque provided by the joint actuator, *l*_4_ is the length of the tail link, and *m*_4_ is the mass of the tail link. Note that for the sake of simplicity, we applied a safety factor of 0.6 in (7) and a point mass model for the tail link. The mass of arm link *m*_3_ lumped together with the enclosed actuator can be estimated as follows:(8)$$m_{3}=bl_{3}+m_{a}$$
where *b* indicates the ratio of mass to length in the arm link (2.1 kg/m) and *m_a_* indicates the mass of the actuator used to manipulate the arm link (0.077 kg).

Modeling of the tail link is simplified as a point mass pendulum comprising a tail actuator with electronic components for motion control and power delivery. The length of the tail link could be extended to increase swing momentum; however, this would impose a corresponding increase in joint loading. For the sake of simplicity, we assumed that tail length *l*_4_ is half the total robot length, as shown in Figure 5. As shown in Figure 6, the resulting mass *m*_4_ is approximately 0.5 kg. Note that the design of the gripper does not depend on the specifications of the other links, which means that it can be designed and fabricated as an independent module. The mass of the gripper used in this study was 0.14 kg. Applying these specifications to (7), we determined that *l*_3_ should be ≤0.1074 m. We eventually selected an *l*_3_ value of 0.102 m.

## 3. Robot System Integration

### 3.1. Robot Mechanism

Our analysis of humans traversing ledges suggested a robot design that includes an upper body comprising two arms with wrist joints and a lower body comprising only a tail. Figure 7 presents a 3D CAD drawing of the proposed transverse brachiation robot. 

As shown in Figure 7, the mechanical transmission system includes wrist joints, arm joints, and a tail joint. We employed passive wrist joints to facilitate the accumulation of kinetic energy during the swing phase. Note, however, that passive wrist joints render the gripper susceptible to rotational misalignment with the target ledge. We resolved this problem by incorporating a belt mechanism in the wrist joints to enable the gripper of the grasping hand to maintain the same wrist posture as the gripper of the support hand. Actuators transmit power through belt mechanisms to arm joints. An actuator in the tail joint is used to introduce a swinging motion via bevel gears. A slip ring device installed in the body link enables the efficient transmission of power used for tail rotation. 

### 3.2. Gripper for Transverse Ledge Brachiation

Unlike bars and cables, the target for ledge brachiation robots is flat. Thus, the gripper should provide rapid open and close operations to enable the clamping of the rail-like ledge. It is also important that the claws do not strike the target ledge prior to the grasping phase. A 3D CAD drawing of the proposed gripper system is presented in Figure 8. The gripper comprises three parts: Upper claw, lower claw, and pulley system. The upper claw comprises a pair of support blocks attached to the gripper base via vertical hinges to enable horizontal rotation via a servo motor. A torsion spring integrated into the hinge stores elastic energy to enable the rapid closing of the upper claw, and a locking mechanism is used to keep the upper claw open until triggered, as shown in Figure 8b. The lower claw mechanism includes components that independently control the opening and closing of the two claws using a single servo motor comprising a drive disk and two control disks, as shown in Figure 8c. Power is transferred from the motor to the claws using the drive disk. The control disk attached to the lower claw drives the closing action of the upper claws as well as the clamping action of the lower claw. The other control disk drives the opening of the upper claws. In accordance with the concept of return difference or backlash, the direction of drive disc rotation is used to determine the state of the two control disks in order to apply independent actuations by which to control the moving parts of the upper and lower claws. As shown in Figure 8d, two control cables in the pulley mechanism open and close the upper claws, wherein the control string connected to the control disk of the lower claw triggers the device to close, whereas the control cable connected to the control disk of the upper claws triggers the device to open.

### 3.3. Embedded Control System

Motion control is implemented using an embedded control platform (STM32F407 discovery board, STMicroelectronics, Geneva, Switzerland), a schematic diagram of which is shown in Figure 9. The tail joint is actuated by a brushed DC motor (DCX22L, Maxon, Switzerland) in conjunction with a motor driver (HB-25, Parallax, Rocklin, CA, USA), and tail angle feedback is obtained using a quadrature decoder (HCTL-2032, Avago, San Jose, CA, USA). A PID (Proportional Integral Derivative) feedback control system performs tail motion control. Two servomotors (MX-28AT, Dynamixel, Lake Forest, CA, USA) installed in the arm joints control the movement of the arm links via TTL serial communication. Two inertial measurement units (IMUs) (GY953, Huimai, Shanghai, China) attached to the arm links are used to determine the absolute joint angle in accordance with the SPI communication protocol. All experiment data were stored in an SD card and transmitted wirelessly to a computer for analysis.

## 4. Robot Modeling

In this section, we present a dynamic model of the robot, which takes into account the dynamics of the joint actuator and the effects of the backlash caused by reduction gears in the tail actuator. We use this model in simulations by which to analyze design parameters.

As shown in Figure 2, robot locomotion begins with the release of the orange gripper, while the robot is suspended like a pendulum beneath the blue gripper. As shown in Figure 10, our four-link dynamic model has all of the mechanical parts and actuators lumped together with the arms, body, and tail as a single unit under the assumption that there is no relative motion between the support ledge and support gripper. In other words, the robot swings freely beneath the support hand. The center point of the support gripper where it rests on the support ledge defines the origin of the coordinate system. The physical meaning of the parameters in Figure 10 is summarized in Table 2.

The Lagrangian ℒ can be defined as follows
(9)$$ℒ=E_{robot}-V_{robot}$$
where *E_robot_* and *V_robot_*, respectively, indicate the kinetic energy and potential energy of the robot based on the coordinates defined in Figure 10. The dynamic equation can be obtained using the Lagrangian method as follows:(10)$$\left\{ \begin{array}{l} \frac{d}{dt}(\frac{\partial ℒ}{\partial {\dot{\theta }}_{w}})-(\frac{\partial ℒ}{\partial \theta_{w}})=-d_{w}{\dot{\theta }}_{w}\\ \frac{d}{dt}(\frac{\partial ℒ}{\partial {\dot{\theta }}_{hb}})-(\frac{\partial ℒ}{\partial \theta_{hb}})=\tau_{h}-d_{h}{\dot{\theta }}_{hb}\\ \frac{d}{dt}(\frac{\partial ℒ}{\partial {\dot{\theta }}_{br}})-(\frac{\partial ℒ}{\partial \theta_{br}})=\tau_{r}-d_{r}{\dot{\theta }}_{br}\\ \frac{d}{dt}(\frac{\partial ℒ}{\partial {\dot{\theta }}_{bt}})-(\frac{\partial ℒ}{\partial \theta_{bt}})=\tau_{t}-d_{t}{\dot{\theta }}_{bt} \end{array}\right.$$

Simplifying (10) and skipping the derivation details gives the following standard model of robot dynamics:(11)M(q)q¨+C(q,q˙)+G(q)+F=Bu
where M(q)∈ℝ4×4 is the inertia matrix, C(q,q˙)∈ℝ4×1 is the coriolis/centrifugal matrix, G(q)∈ℝ4×1 is the gravity matrix, and q=[θw,θhb,θbr,θbt]T denotes the four states of the robot. F∈ℝ4×1 refers to the friction force vector, and *Bu* refers to the actuator torque vector, where
F=[dwθ˙w000], B=[000100010001],u=[τhτrτt]

Note that the damping coefficients of the arms and tail link are absent here, and the damping effects are included in the actuator dynamics that generate torques *τ_h_*, *τ_r_*, and *τ_t_*. Due to the non-negligible backlash effects of the gears in the tail actuator transmission [36], the torque of the tail joint *τ_t_* can be re-written as follows:(12)$$\left\{ \begin{array}{l} \tau_{t}=k\left(\theta_{mt}-\theta_{t}-h_{t}\right)+c\left({\dot{\theta }}_{mt}-{\dot{\theta }}_{t}\right)\;\mathrm{when}\;\left(\theta_{mt}-\theta_{t}\right)>h_{t}\\ \tau_{t}=0\;\;\;\;\;\;\;\;\;\;\;\;\;\;\;\;\;\;\;\;\;\;\;\;\;\;\;\;\;\;\;\;\;\;\;\;\mathrm{when}\;\left|\theta_{mt}-\theta_{t}\right|\leq h_{t}\\ \tau_{t}=k\left(\theta_{mt}-\theta_{t}+h_{t}\right)+c\left({\dot{\theta }}_{mt}-{\dot{\theta }}_{t}\right)\;\mathrm{when}\;\left(\theta_{mt}-\theta_{t}\right)<-h_{t} \end{array}\right.$$
where $\theta_{mt}$ and ${\dot{\theta }}_{mt}$ respectively refer to the output angle and angular velocity of the tail actuator, *h_t_* is the backlash caused by the transmission between tail link and actuator, and *k* and *c* represent the parameters used in modeling the contact dynamics of the gear teeth (stiffness and damping coefficient). The dynamic equation of the tail actuator combining DC motor dynamics can be represented as
(13)$$\left(\eta N^{2}J_{m}{\ddot{\theta }}_{mt}+\eta NC_{m}\mathrm{sgn}(N{\dot{\theta }}_{mt})+\eta N^{2}\frac{K_{T}K_{b}}{R_{a}}{\dot{\theta }}_{mt}\right)+\tau_{t}=\eta N\frac{K_{T}}{R_{a}}v_{t}$$
where *J_m_* is the rotor moment of inertia, *C_m_* is the frictional coefficient, *K_T_* and *K_b_* are the motor constants, *R_a_* is the armature resistance, *N* is the gear ratio, *η* is the gear efficiency, and *v_t_* is the motor voltage input. Similarly, the dynamic equation of the joint actuators for support arm and releasing arm can be derived as
(14)$$\left\{ \begin{array}{l} \tau_{h}=\eta_{s}N_{s}\frac{K_{T}}{R_{a}}v_{h}-\left(\eta_{s}N_{s}^{2}J_{s}{\ddot{\theta }}_{h}+\eta_{s}N_{s}C_{s}\mathrm{sgn}(N_{s}{\dot{\theta }}_{h})+\eta_{s}N_{s}{}^{2}\frac{K_{T}K_{b}}{R_{a}}{\dot{\theta }}_{h}\right)\\ \tau_{r}=\eta_{s}N_{s}\frac{K_{T}}{R_{a}}v_{r}-\left(\eta_{s}N_{s}^{2}J_{s}{\ddot{\theta }}_{r}+\eta_{s}N_{s}C_{s}\mathrm{sgn}(N_{s}{\dot{\theta }}_{r})+\eta_{s}N_{s}{}^{2}\frac{K_{T}K_{b}}{R_{a}}{\dot{\theta }}_{r}\right) \end{array}\right.$$
Here the two actuators share the same DC motor parameters, where subscript notation *h* and *r* are used to distinguish between two arm actuators, where *θ_h_* and *θ_r_* represent the output angles obtained from encoders.

## 5. Locomotion Control for Transverse Ledge Brachiation

As defined in Section 2, the phases of the proposed robot locomotion include: (1) release phase, (2) body reversal phase, (3) swing-up phase, and (4) grasping phase. In this section, we describe the motion control strategy used in each phase and the switching conditions applied to realize smooth phase transitions. 

### 5.1. Control Strategy for Phase 1: Release Phase

The initial posture has the robot suspended below the hands resting on the ledge. Given a desired target ledge grasping position (*x_d_*, *y_d_*), the robot determines the direction of movement and releases one hand from the held ledge in accordance with the following conditions:(15)$$\left\{ \begin{array}{l} \mathrm{release}\;\mathrm{right}\;\mathrm{gripper}\;\mathrm{and}\;\mathrm{move}\;\mathrm{left},\;\mathrm{if}\;x_{d}<x_{l}\\ \mathrm{release}\;\mathrm{left}\;\mathrm{gripper}\;\mathrm{and}\;\mathrm{move}\;\mathrm{right},\;\mathrm{if}\;x_{d}>x_{r} \end{array}\right.$$
where *x_l_* and *x_r_*, respectively, denote the *x*-coordinates of the left and right grippers. In this phase, the robot is able to lift the tail link to increase the potential energy and reduce the swing time in subsequent phases. The reference command controlling the angle between the body link and tail link, $\theta_{bt}^{d}$, is as follows:(16)$$\left\{ \begin{array}{l} \theta_{bt}^{d}=180^{\circ },\;\mathrm{if}\;\mathrm{releasing}\;\mathrm{right}\;\mathrm{gripper}\;\mathrm{and}\;\mathrm{moving}\;\mathrm{left}\\ \theta_{bt}^{d}=0^{\circ },\;\mathrm{if}\;\mathrm{releasing}\;\mathrm{left}\;\mathrm{gripper}\;\mathrm{and}\;\mathrm{moving}\;\mathrm{right} \end{array}\right.$$

The robot switches to the body reversal phase when the velocity of the wrist joint (${\dot{\theta }}_{w}$) is not equal to zero (or exceeds the sensor noise level) following the release of the support hand from the held ledge.

### 5.2. Control Strategy for Phase 2: Body Reversal Phase

Phase 2 is a transition phase designed to adjust the robot’s posture before moving on to the subsequent next phase involving overhand swing motion. The idea is to regulate the motion of the arm joint using angle commands $\theta_{hb}^{d}$ and $\theta_{br}^{d}$ and realize a simplified two-link robot posture by tracking the trajectory of the tail joint $\theta_{bt}^{d}$. The equations used to plan the trajectories of the joint are as follows:(17)$$\left\{ \begin{array}{l} \theta_{hb}^{d}=\left(1-b_{1}\right)\theta_{hb}^{s}+b_{1}\theta_{hb}^{f}\\ \theta_{br}^{d}=\left(1-b_{1}\right)\theta_{br}^{s}+b_{1}\theta_{br}^{f}\\ \theta_{bt}^{d}=2\alpha /\pi \left[\tan^{-1}({\dot{\theta }}_{w})\right]+\theta_{offset} \end{array}\right.$$

The values of $\theta_{hb}^{f}$ and $\theta_{br}^{f}$ can be derived using Equation (3) with the target ledge position (*x_d_*, *y_d_*). $\theta_{hb}^{s}$ and $\theta_{br}^{s}$ indicate the arm joint angles obtained at the moment the robot transfers to the body reversal phase; *b*_1_ is a time-varying translational coefficient [37] used to alleviate transient impact and energy loss during body posture adjustment, which can be expressed as follows:(18)$$b_{1}=\left\{ \begin{array}{l} 0.5\left[1-\cos(\pi t_{1}/T_{1})\right],\left(t_{1}<T_{1}\right)\\ 1,\left(t_{1}\geq T_{1}\right) \end{array}\right.$$
where *t*_1_ refers to the elapsed time since the initiation of the body reversal phase, and *T*_1_ is the time period derived in phase 2. In Equation (17), we applied an energy bumping method [32] to generate the tracking trajectory of the tail joint ($\theta_{bt}^{d}$), based on information related to wrist joint velocity
${\dot{\theta }}_{w}$. It is crucial that we coordinate the moving direction of the tail link and robot moving direction; therefore, we included parameter *θ_offset_* from (19) in (17) to correct the tail swing motion and facilitate the ledge grasping action in subsequent phases.
(19)$$\left\{ \begin{array}{l} \theta_{offset}=90^{\circ },\;\mathrm{if}\;\mathrm{releasing}\;\mathrm{right}\;\mathrm{gripper}\;\mathrm{and}\;\mathrm{moving}\;\mathrm{left}\\ \theta_{offset}=-90^{\circ },\;\mathrm{if}\;\mathrm{releasing}\;\mathrm{left}\;\mathrm{gripper}\;\mathrm{and}\;\mathrm{moving}\;\mathrm{right} \end{array}\right.$$

### 5.3. Control Strategy for Phase 3: Swing-Up Phase

In the swing-up phase, we gave the robot a simplified two-link posture with the two arm joints maintained at angles $\theta_{hb}^{f}$ and $\theta_{br}^{f}$ during the tail swing and gradually increased the swing amplitude. The joint motion strategy in this phase is based on (17), except that here *b*_1_ = 0. After several swing cycles, the phase switching conditions in (20) are used to determine whether the robot is ready to grasp the target ledge and progress to the following phase. Consider the case in Figure 11, where the left gripper is released, and the robot moves to the right toward the target position (*x_d_*, *y_d_*). The ideal wrist joint angle $\theta_{w}^{f}$ for grasping can be derived using (21). The objective is to determine whether the maximum wrist joint angle $\theta_{w}^{k}$ has attained a threshold value $a\theta_{w}^{f}$, whereupon further action is delayed for one cycle to maximize wrist joint velocity, ${\dot{\theta }}_{w}^{k+1}$. Note that a>1 is a safety coefficient used to compensate for potential energy loss in the following phase.
(20)$$\begin{array}{l} \left\{ \begin{array}{l} \min\left(\theta_{w}^{k}\right)\leq a\theta_{w}^{f}\;\&\;{\dot{\theta }}_{w}^{k+1}\to \min\left({\dot{\theta }}_{w}^{k+1}\right),\;\mathrm{if}\;\mathrm{releasing}\;\mathrm{right}\;\mathrm{gripper}\;\mathrm{and}\;\mathrm{moving}\;\mathrm{left}\\ \max\left(\theta_{w}^{k}\right)\geq a\theta_{w}^{f}\;\&\;{\dot{\theta }}_{w}^{k+1}\to \max\left({\dot{\theta }}_{w}^{k+1}\right),\;\mathrm{if}\;\mathrm{releasing}\;\mathrm{left}\;\mathrm{gripper}\;\mathrm{and}\;\mathrm{moving}\;\mathrm{right} \end{array}\right.\\ k:\;\mathrm{current}\;\mathrm{swing}\;\mathrm{cycle},\;k+1:\;\mathrm{next}\;\mathrm{swing}\;\mathrm{cycle} \end{array}$$
(21)$$\theta_{w}^{f}=\frac{\pi }{2}+\tan^{-1}(\frac{y_{d}}{x_{d}})-\theta_{hb}^{f}$$

### 5.4. Control Strategy for Phase 4: Grasping Phase

There is an inherent delay between the time that a command is sent and the time at which the gripper responds, and the grasping action cannot be completed after the support gripper passes beyond the target ledge. The primary objective here is to complete the grasping action within a short time window (i.e., without missing the target ledge), while maintaining a posture suitable for grasping. We sought to resolve this problem by sending grasping commands slightly before the anticipated arrival time.

During this phase, the tail link ceases its active swing motion, such that the robot performs a free-swinging motion toward the target under the effects of inertia. During this period, it is possible to predict the wrist joint swing angle by approximating the swing trajectory for ${\hat{\theta }}_{w}$ as a sinusoidal profile. The conditions under which the robot sends gripper commands and performs grasping actions are as follows:(22)$$\left\{ \begin{array}{l} {\hat{\theta }}_{w}\leq a\theta_{w}^{f},\;\mathrm{if}\;\mathrm{releasing}\;\mathrm{right}\;\mathrm{gripper}\;\mathrm{and}\;\mathrm{moving}\;\mathrm{left}\\ {\hat{\theta }}_{w}\geq a\theta_{w}^{f},\;\mathrm{if}\;\mathrm{releasing}\;\mathrm{left}\;\mathrm{gripper}\;\mathrm{and}\;\mathrm{moving}\;\mathrm{right} \end{array}\right.$$

The approximated swing trajectory for ${\hat{\theta }}_{w}$ is represented as a sinusoidal function ${\theta^{\prime }}_{w}$ with three parameters derived using joint information from the previous phase: amplitude *A*, offset *B*, and frequency *f*. The equations used to derive ${\theta^{\prime }}_{w}(t)$ and ${\dot{\theta^{\prime }}}_{w}(t)$ are as follows:(23)$$\left\{ \begin{array}{l} {\theta^{\prime }}_{w}(t)=A\sin(2\pi ft)+B\\ {{\dot{\theta }}^{\prime }}_{w}(t)=2\pi fA\cos(2\pi ft) \end{array}\right.$$
(24)$$A=\left[\max(\theta_{w}^{S})-\min(\theta_{w}^{S})\right]/2$$
(25)$$\left\{ \begin{array}{l} B=\min(\theta_{w}^{S})-A,\;\mathrm{if}\;\mathrm{releasing}\;\mathrm{right}\;\mathrm{gripper}\;\mathrm{and}\;\mathrm{moving}\;\mathrm{left}\\ B=\max(\theta_{w}^{S})-A,\;\mathrm{if}\;\mathrm{releasing}\;\mathrm{left}\;\mathrm{gripper}\;\mathrm{and}\;\mathrm{moving}\;\mathrm{right} \end{array}\right.$$

Parameter *A* is half the amplitude of the previous swing motion, expressed as (24). Parameter *B* is based on the direction of robot movement using (25), where superscript *S* is used to indicate that the wrist joint variables were acquired during the swing-up phase.

Swing frequency *f* can be derived by setting the amplitude of the swing velocity to half the velocity range in the previous swing motion, expressed as follows:(26)$$2\pi fA=\left[\max({\dot{\theta }}_{w}^{S})-\min({\dot{\theta }}_{w}^{S})\right]/2$$
Substituting (24) into (26) gives us a reasonable estimate of the swing trajectory (23).
(27)$$f=\frac{\max({\dot{\theta }}_{w}^{S})-\min({\dot{\theta }}_{w}^{S})}{2\pi \left[\max(\theta_{w}^{S})-\min(\theta_{w}^{S})\right]}$$

Given the current wrist joint angle *θ_w_*, we use timing *t* to predict swing angle ${\hat{\theta }}_{w}$ at the next moment based on (28). The predicted swing trajectory used to trigger the gripper action in (22) is derived using (29).
(28)$$t=\sin^{-1}(\frac{\theta_{w}^{S}-B}{A})/2\pi f$$
(29)$${\hat{\theta }}_{w}(t)=A\sin(2\pi f(t+\Delta t))+B$$
(30)$$\theta_{w}\approx \theta_{w}^{f}\;\&\;{\dot{\theta }}_{w}\approx 0$$
where Δ*t* indicates the time delay after receiving the command to initiate gripper motion, wherein the point at which the gripper system is actually activated is estimated or measured offline. When grasping Condition (22) is met, the robot determines whether the grasping action has been successful or failed, based on Condition (30). A threshold value is applied to (30) to deal with measurement noises and numerical issues. If Condition (30) is TRUE, then one locomotion cycle has been completed; otherwise, the motion control strategy iteratively returns to phase 3 to try again.

### 5.5. Control Strategy: Summary

The motion control strategy used in the previous four phases is summarized using the integrated control block diagram in Figure 12, where $\theta_{hb}^{d},\theta_{br}^{d},\theta_{bt}^{d}$ are the calculated reference commands sent to robot joint actuators and $\theta_{w},{\dot{\theta }}_{w},\theta_{hb},\theta_{br},\theta_{bt}$ are feedback information from the encoder and IMU sensors. Five switchable states are used to represent the phases in each locomotion cycle. State 1 refers to the release phase, during which the robot releases one hand from the support ledge and switches to State 2 when the velocity change in the wrist joint reaches a threshold. State 2 refers to the body reversal phase, during which joint motion is adjusted to accommodate the simplified two-link posture used to perform the swing motion. Note that posture adjustment must be completed within a set time period (*T*_1_). State 3 refers to the swing-up phase, during which the tail link is swung in order to accumulate kinetic energy sufficient to propel the non-support gripper to an elevation exceeding that of the target ledge while preparing to grasp the target ledge in the following phase based on Condition (12). State 4 refers to the grasping phase, during which the timing used to compensate for a delay in the gripper system is calculated. The locomotion cycle is completed at the point where the robot succeeds in grabbing the target ledge; i.e., Condition (30) is satisfied. Note that if the robot fails to grab the target ledge, the system reverts to State 3 and repeats the process. State 5 refers to the end of the process.

### 5.6. Simulations

We assessed the feasibility of the proposed locomotion and control strategy by conducting computer simulations involving level ledge brachiation with movement toward the right. The numerical values of model parameters were obtained using CAD software (see Table 3). The parameters of the DC motors were obtained using a system identification process [8].

Figure 13 illustrates the initial posture of the robot prior to horizontal ledge brachiation, in which the target ledge is located at (0.2 m, 0 m). The robot in the animated figure releases the left hand (at −0.2 m, 0 m) and then goes through the motions involved in reaching the target ledge based on the locomotion control strategy in Figure 12. The simulation results of the three subsequent phases are presented in Figure 14. We applied a proportional control law with parameter *K_p_* = 20 to control the tail swing motion. We calculated the ideal wrist joint angle ($\theta_{w}^{f}$) for grasping as 40.69°, based on (21). Application of the compensation coefficient (*a* = 1.05) and $\theta_{w}^{f}$ resulted in a threshold value of 42.72°. As shown in Figure 14, during the swing-up phase, the maximum wrist joint angle in the first swing cycle (at 1.453 s) was 45.11°, which exceeded the threshold value (42.72°). In accordance with Condition (20), the robot initiated the grasping phase in the following cycle at the moment the maximum wrist joint velocity was attained during its swing toward the target ledge.

Using (29), while taking into account the gripper time delay (Δ*t* = 0.1 s), we determined that the predicted wrist joint angle ${\hat{\theta }}_{w}$ would exceed the threshold value when ${\hat{\theta }}_{w}=42.73{}^{\circ}$ (at 2.234 s). As shown in Figure 14, a command was sent to the gripper to initiate the ledge grasping sequence at 2.234 s, as the robot swung freely with a fixed tail joint angle *θ_t_* (≈−49°). Figure 14b compares this value to the wrist joint motion derived from the predicted swing trajectory (29) in the simulation. The wrist joint angle derived via modeling (*θ_w_* = 41.53°) was slightly less than the predicted angle ${\hat{\theta }}_{w}=42.73{}^{\circ}$; however, the difference was well within an acceptable range and could be eliminated simply by increasing compensation coefficient *a* in (22).

## 6. Experiments and Discussion

The efficacy of the proposed robot design and locomotion control strategy was evaluated in two sets of experiments: horizontal ledge brachiation and non-horizontal-elevation ledge brachiation. The testing scenarios examined in this study focused on horizontal parallel ledges, i.e., no variation in the pitch angle of any ledges.

We constructed an experiment test bed using 20 mm × 20 mm extruded aluminum bars supporting white plastic pedestals for the hands. In the following six experiments, the support gripper was secured on the ledge using a C-clamp to preserve the boundary conditions. Note that these initial experiments involved horizontal ledge brachiation, and the control law used for tail motion control was the same as that used in the simulations. Our primary objective here was to assess the effects of gripper time delay on ledge grasping performance. We then applied the proposed locomotion control strategy in experiments involving non-horizontal-elevation brachiation.

### 6.1. Experiment: Horizontal Ledge Brachiation

Figure 15 illustrates the setup used in the first set of transverse ledge brachiation experiments. Note that the ledges are at the same elevation, separated by a gap of 210 mm. The experiment results are listed in Table 4, and the joint motion is illustrated in Figure 16, Figure 17, Figure 18, Figure 19, Figure 20 and Figure 21. We implemented two strategies for the activation of gripper commands. The first strategy involved sending gripper commands based solely on information pertaining to the current robot swing motion (Experiments (a), (b), and (c)). The second strategy involved sending gripper commands using the proposed time delay (Experiments (d), (e), and (f)).

In Experiment (a), the gripper was triggered when the robot swing angle *θ_w_* reached $\theta_{w}^{f}$$\theta_{w}^{f}$ (≈45°); i.e., precisely between the swing-up phase to grasping phase (see Figure 16). As shown in Table 4, this simple strategy resulted in brachiation failure, with the robot falling to the ground. In the following experiment, we adjusted the timing of the maximum swing magnitude during the swing-up phase (≈48.05° at 8.245 s) by imposing a time delay in triggering the gripper (slightly greater than the value used in simulation, approximately 0.14 s). The resulting magnitude was ≈32° at 8.1 s. As shown in Figure 17, the robot activated the gripper at 9.1 s (i.e., when the swing magnitude reached the threshold value of 31.98°). Note that the numerical values indicated in the plots are for reference only and may differ somewhat from those in Table 4. 

The strategy employed in Exp. (b) achieved in brachiation success, with a total execution time of 9.62 s. Note, however, that when this experiment was repeated, we observed a number of failures (see Experiment (c) in Figure 18). This lack of robustness could perhaps be attributed to perturbations in wrist joint motion during the swing phase, which can, in turn, be attributed to nonlinear effects, such as gripper contact dynamics or time-varying friction. Nonlinear effects would no doubt skew the magnitude of sinusoidal motion.

We sought to enhance the robustness of the system by applying an approximated sinusoidal function (29) using parameters based on historical results of wrist joint motion during the swing-up phase. All three trials using the triggering condition in Equation (22) achieved successful outcomes, even under the effects of disturbance (see Figure 19, Figure 20 and Figure 21). Note that the maximum swing magnitude and swing duration varied between the three trials. For example, the duration of the swing-up phase in Experiment (e) was 13.46 s, far exceeding that in Exp. (d) (10.32 s). Note, however, that increasing the swing time did not increase the swing magnitude: Exp. (e) (57.46°) and Exp. (f) (56.18°). The maximum swing magnitude in Exp. (e) occurred in the middle of the swing-up phase (57.46° at 10.31 s). This can be attributed to the fact that the robot may have a collision with the ledge while approaching the target. This undesired collision makes the robot shake and deviate from the vertical swing plane, increasing the risk of robot collision and energy dissipation. Thus, in Exp. (e) the robot needed to take more time to satisfy the phase switching Condition (20) exceeding the default setting value of $\theta_{w}^{f}$.

Overall, the time delay increased the total execution time for a single locomotion cycle compared to Experiments (a), (b), and (c). This is the price for improving system robustness and robot safety, which is the primary priority in designing a robot to climb on a vertical wall.

### 6.2. Experiment: Non-Horizontal-Elevation Ledge Brachiation

The first set of experiments described above demonstrated the feasibility of the proposed motion control strategy when dealing with a target that is level with the starting point. Our next goal was to assess the feasibility of the proposed motion control strategy when dealing with a target that is higher than the starting point. As shown in Figure 22, we set up an experiment platform that allowed the manual adjustment of the target ledge in terms of gap distance and elevation. In Exp. (g), we increased the distance to the target from 210 mm (previous experiments) to 250 mm. In Exp. (h), the target was set at 56 mm above the origin ledge at a horizontal distance of 192 mm. Table 5 lists the parameter settings, including the desired robot joint angles for grasping, the compensation coefficients, maximum swing angle during the swing-up phase, final position of the grasping gripper, and total execution time.

Figure 23 and Figure 24, respectively, present the joint motion and a snapshot in Exp. (g). Figure 25 and Figure 26, respectively, present the joint motion and a snapshot in Exp. (h). In both cases, the maximum swing angle during the swing-up phase exceeded those in the first set of experiments. As shown in Figure 23b, the grasping command was sent at 5.52 sec.; however, the wrist joint angle *θ_w_* at the moment of grasping was 61.15°, which is slightly less than the optimal value $\theta_{w}^{f}$ = 63.77°. Nonetheless, the discrepancy from $\theta_{w}^{f}$ was relatively small (≈2.62°), and the magnitude of *θ_w_* continued increasing after 5.622 sec. The proposed motion control strategy enabled the robot to take advantage of this inertia to grab the target ledge. 

When targeting a ledge higher than the starting position (Exp. (h)), the desired arm joint angles ($\theta_{hb}^{f}$ and $\theta_{br}^{f}$) were roughly double those required for horizontal brachiation (Exp. (g)). Two-dimensional ledge brachiation requires more energy, increases the likelihood of hitting ledges while swinging, and imposes the need for delicate hand-eye coordination to grasp the target ledge. As shown in Figure 25 and Figure 26, the robot successfully completed non-horizontal-elevation brachiation without having to adjust the maximum swing angle during the swing-up phase; however, three additional cycles were required to deal with the effects of gravity. 

Finally, we consider the final position of the gripper derived using feedback from the joint actuator and forward kinematics. The vertical and horizontal coordinates of the gripper refer to the final elevation and distance moved by the gripper following the completion of the brachiation cycle. We determined that the actual elevation of the robot gripper was lower than the desired elevation, which was likely due to the flexibility of the extruded bars used in the experiment setup. As shown in Table 5, all of the horizontal movements were shorter than the desired grabbing distances. Motion captures from video clips revealed that the gripper actually reached or exceeded the target position; however, it then moved back before settling in its final position, indicating the occurrence of slippage associated with contact dynamics. A reference demo video of experiments can be found at the following web link: https://youtu.be/E-Yqpp-tqVI (accessed on 25 May 2022).

## 7. Conclusions

This paper proposes a bio-inspired transverse ledge brachiation robot with a locomotion control strategy aimed at scenarios involving ledges set at different elevations. We formulated a four-link dynamic model to simulate the brachiation process and verify the efficacy of the gripper control strategy in terms of ledge grasping performance. In experiments, the proposed robot proved highly effective in moving horizontally between ledges set at the same and different elevations separated by arbitrary distances. Note that the above experiment results were limited to one locomotion cycle due to a lack of structural rigidity and the effects of cable winding when body reversal was implemented continuously. In future work, we will focus on continuous ledge brachiation sequences and the dynamic relationship between the gripper and target ledge at the point of impact. 

## Figures and Tables

**Figure 1 sensors-22-04031-f001:**
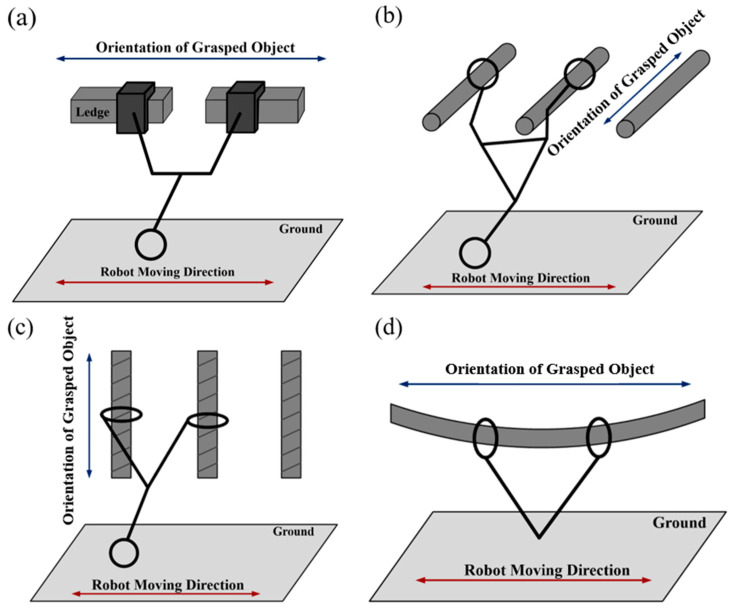
Illustration of principles underlying brachiation robots. (**a**) Proposed robot; (**b**) Gorilla robot III [16]; (**c**) MonkeyBot [23]; (**d**) Tarzan [25].

**Figure 2 sensors-22-04031-f002:**
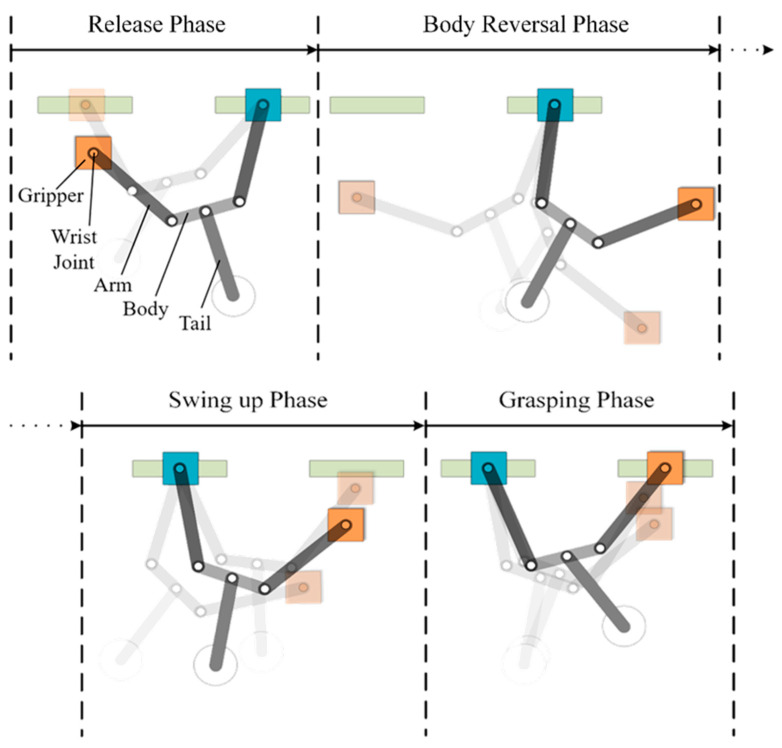
Schematic diagram showing proposed brachiation locomotion scheme.

**Figure 3 sensors-22-04031-f003:**
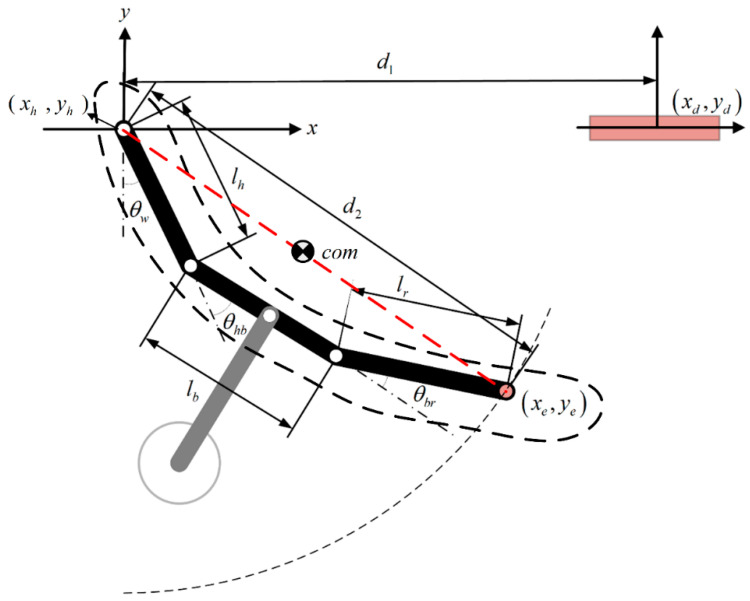
Robot swing trajectory while maintaining a simplified two-link robot posture. The dashed line indicates the trajectory of the lumped link combining the body link and two arm links.

**Figure 4 sensors-22-04031-f004:**
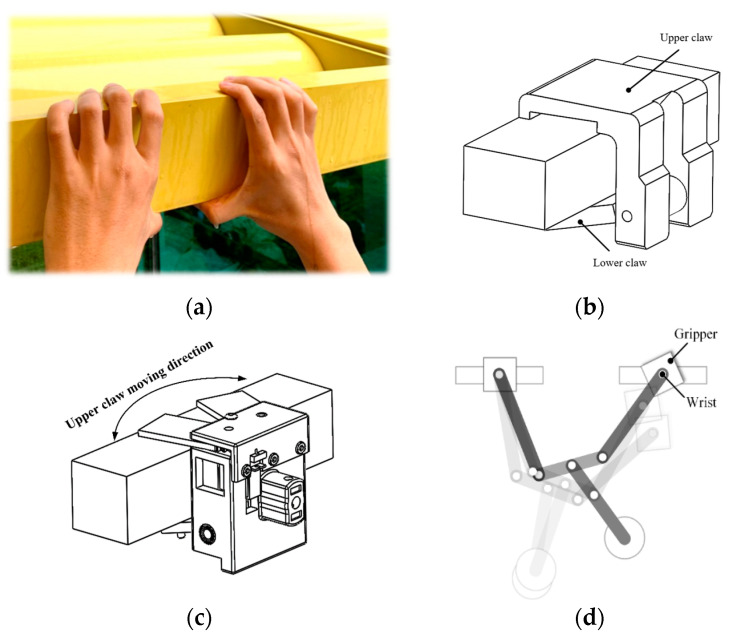
Conceptual design of gripper mechanism: (**a**) athlete supporting body weight on a ledge; (**b**) simplified design; (**c**) proposed design; (**d**) wrist posture deviation.

**Figure 5 sensors-22-04031-f005:**
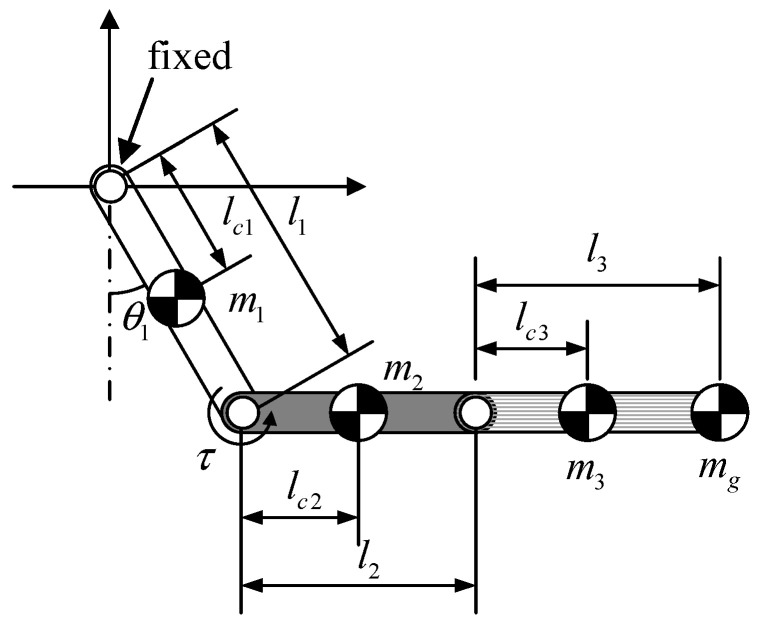
Design parameters with the robot modeled as a three-link system.

**Figure 6 sensors-22-04031-f006:**
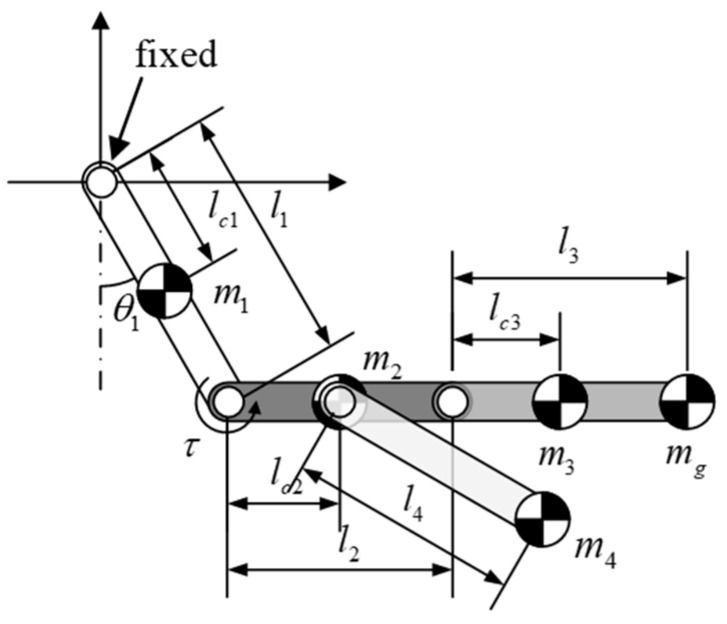
Design parameters with the robot modeled as a four-link system.

**Figure 7 sensors-22-04031-f007:**
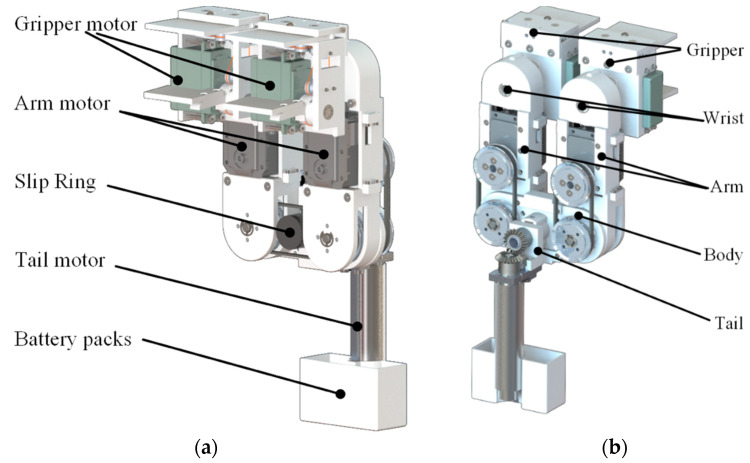
Three-dimensional CAD drawing of proposed transverse brachiation robot: (**a**) front view; (**b**) back view.

**Figure 8 sensors-22-04031-f008:**
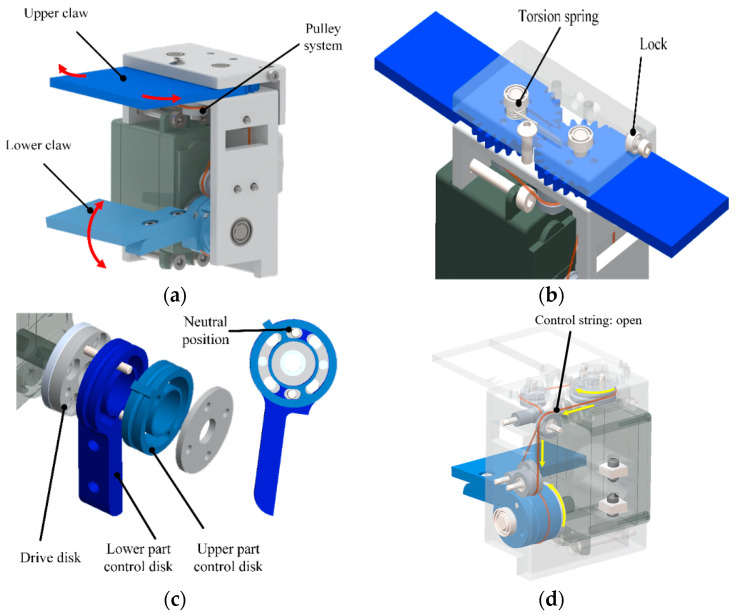
Three-dimensional CAD drawing of proposed gripper: (**a**) overall system; (**b**) upper claw; (**c**) lower claw; (**d**) control string to open the claws.

**Figure 9 sensors-22-04031-f009:**
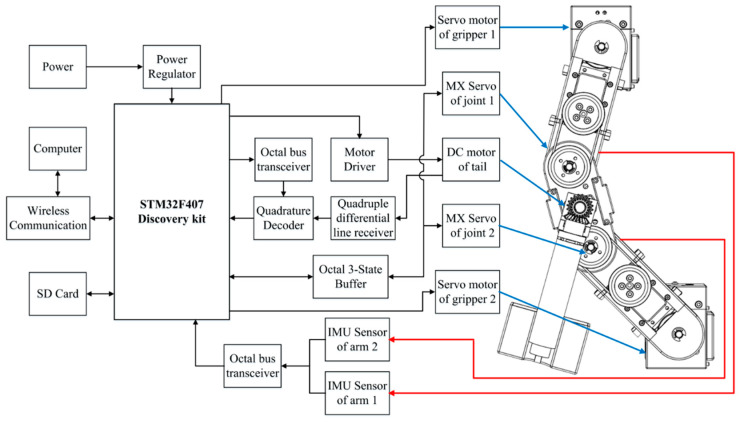
Hardware used in the implementation of the robotic control system.

**Figure 10 sensors-22-04031-f010:**
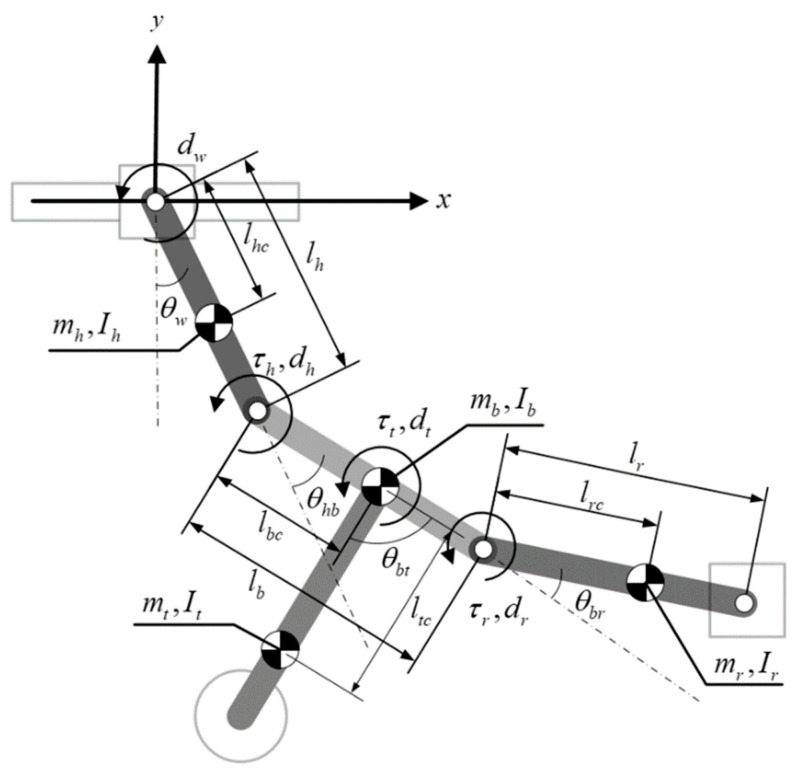
Four-link dynamic model of proposed robot.

**Figure 11 sensors-22-04031-f011:**
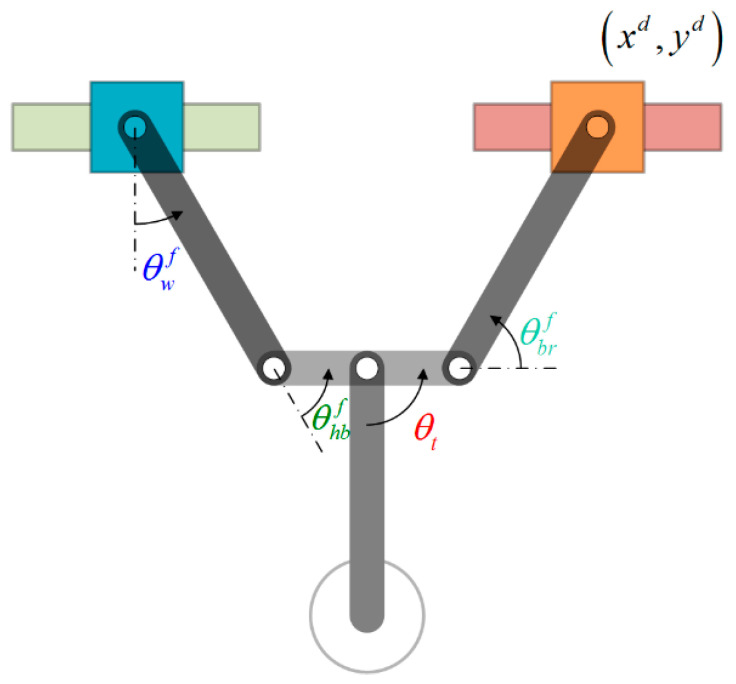
Robot posture best suited to grasping the target ledge.

**Figure 12 sensors-22-04031-f012:**
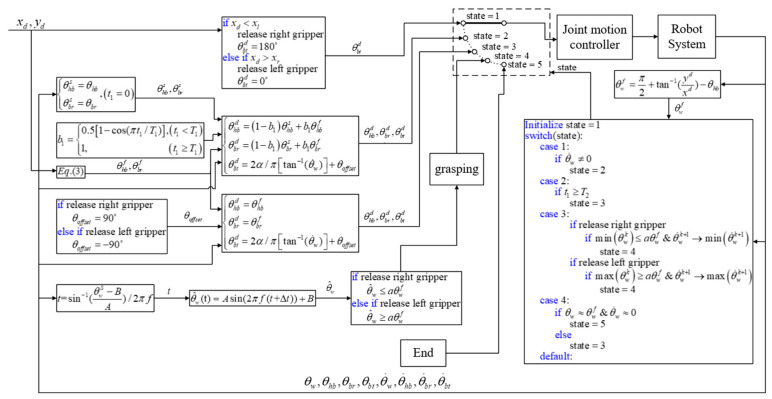
Block diagram showing the proposed locomotion control strategy for transverse ledge brachiation.

**Figure 13 sensors-22-04031-f013:**
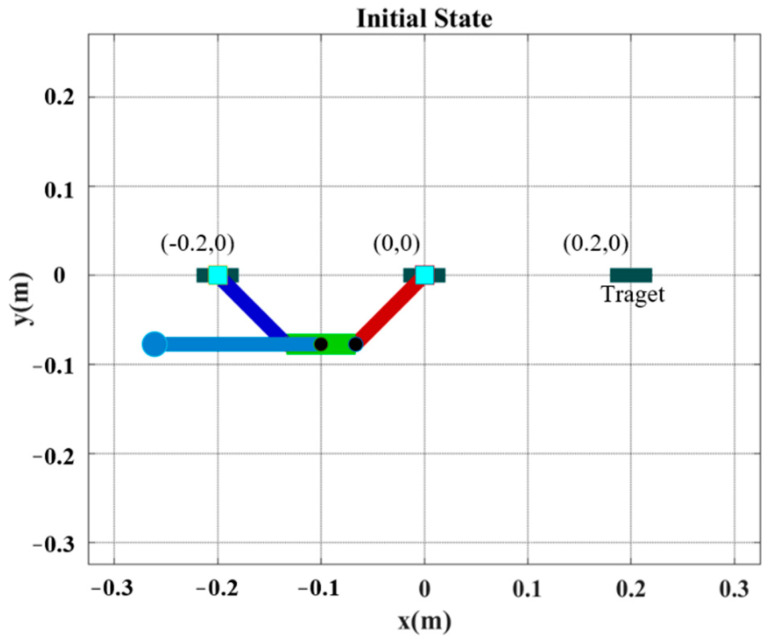
Simulation of ledge brachiation robot: initial posture.

**Figure 14 sensors-22-04031-f014:**
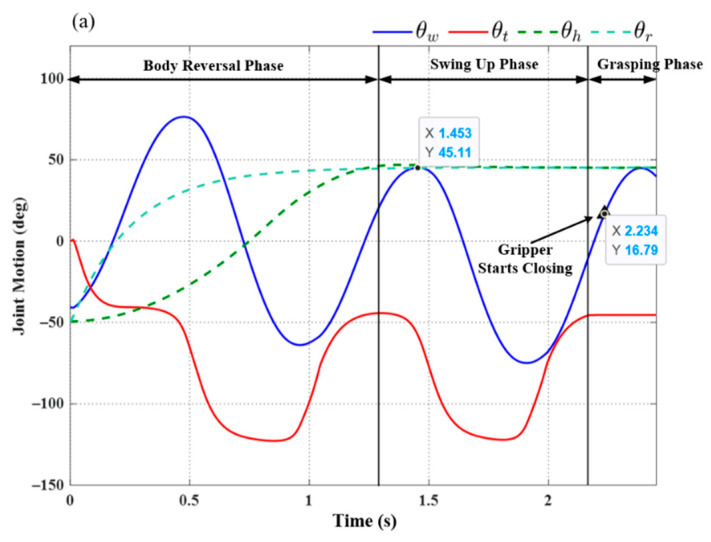
Simulation results: (**a**) joint motion in each phase; (**b**) comparison of wrist joint motions obtained using Equation (29) versus simulation results.

**Figure 15 sensors-22-04031-f015:**
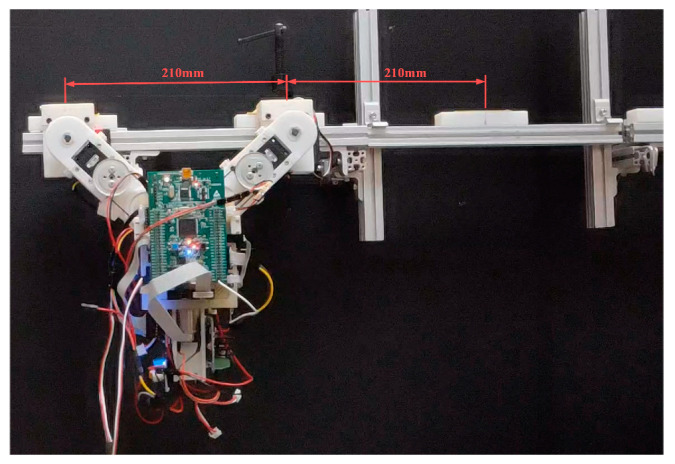
Setup used in the first set of experiments.

**Figure 16 sensors-22-04031-f016:**
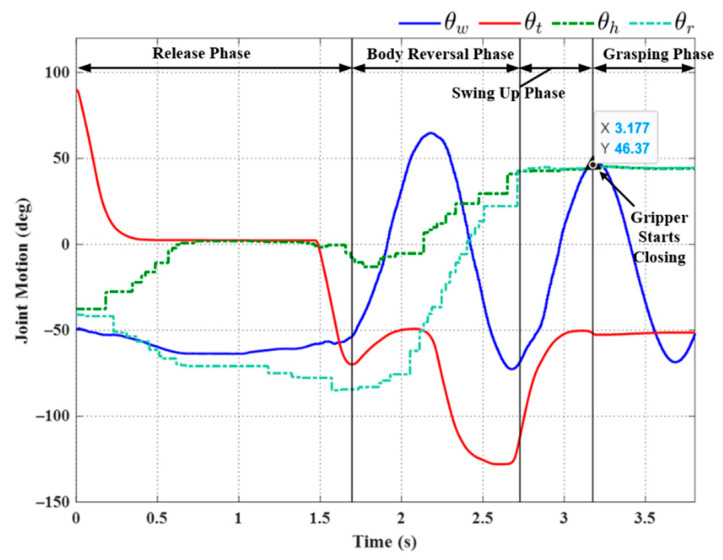
Experimental results: joint motion in Experiment (a).

**Figure 17 sensors-22-04031-f017:**
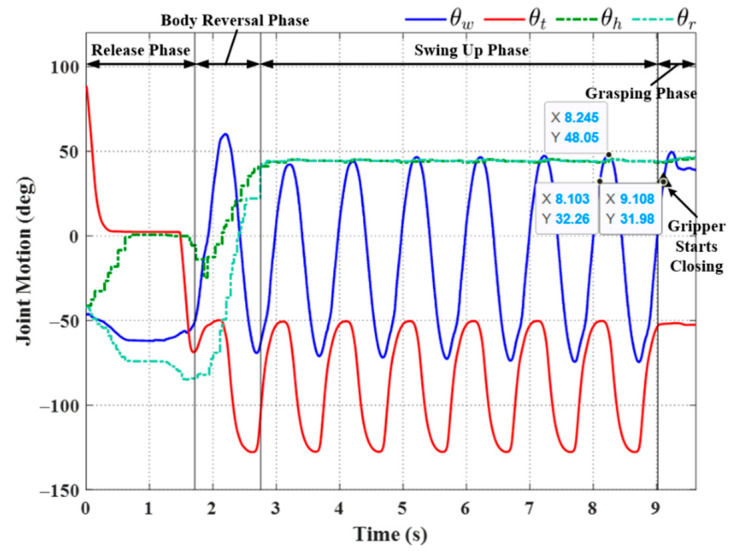
Experimental results: joint motion in Experiment (b).

**Figure 18 sensors-22-04031-f018:**
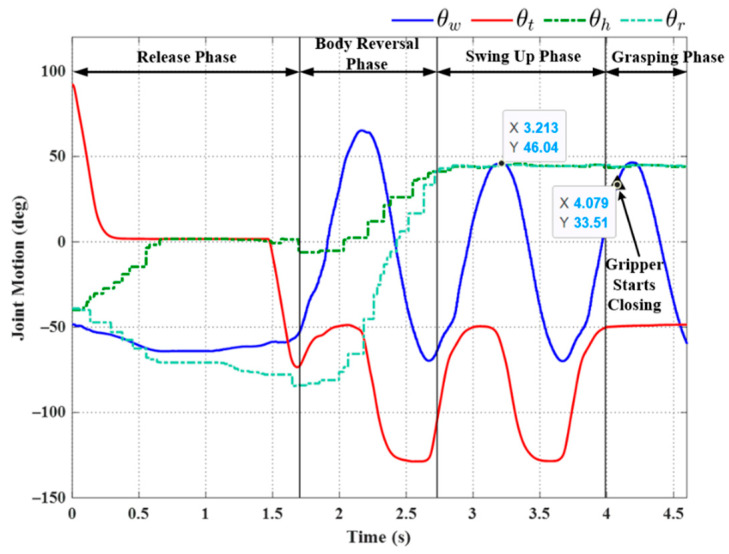
Experimental results: joint motion in Experiment (c).

**Figure 19 sensors-22-04031-f019:**
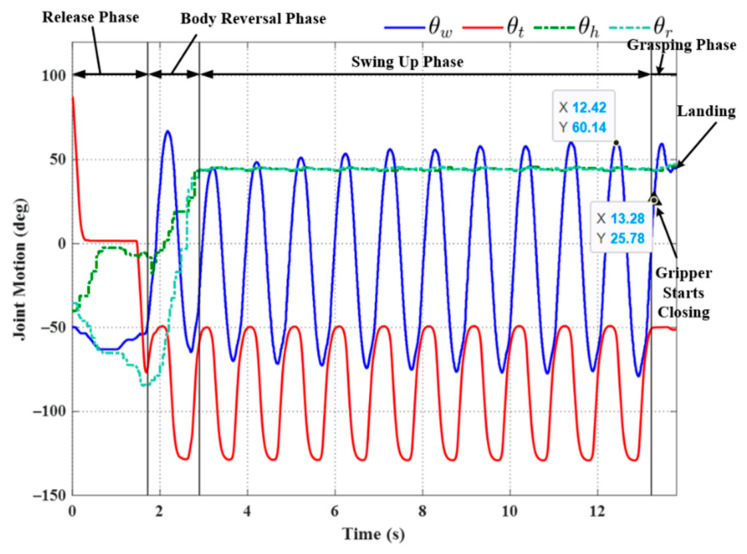
Experimental results: joint motion in Experiment (d).

**Figure 20 sensors-22-04031-f020:**
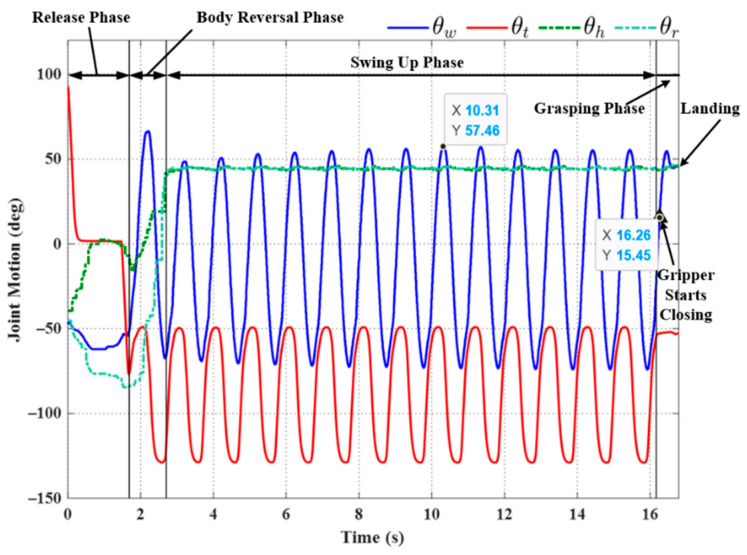
Experimental results: joint motion in Experiment (e).

**Figure 21 sensors-22-04031-f021:**
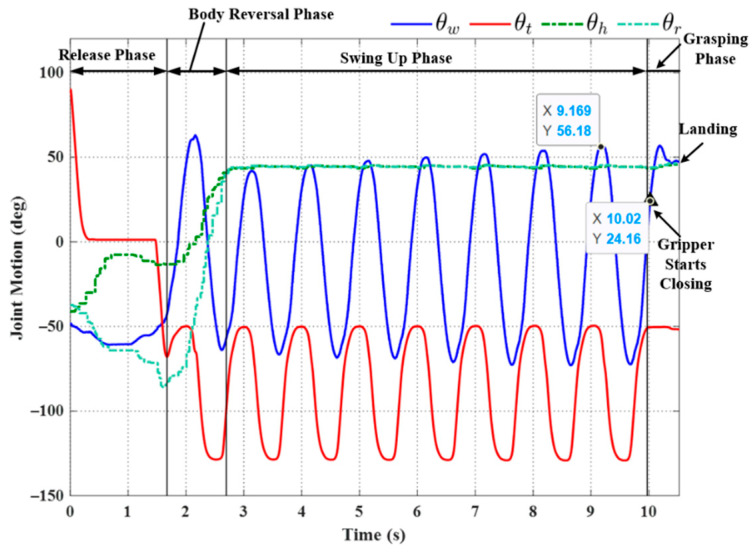
Experimental results: joint motion in Experiment (f).

**Figure 22 sensors-22-04031-f022:**
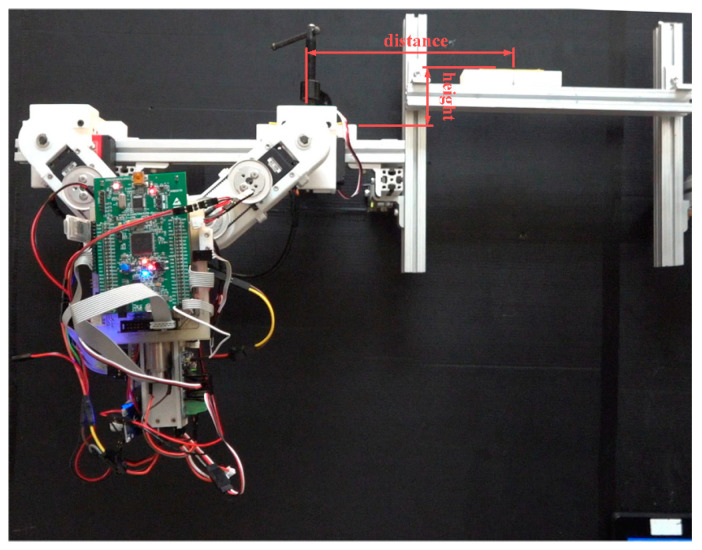
Setup used in second set of experiments: non-horizontal-elevation ledge brachiation.

**Figure 23 sensors-22-04031-f023:**
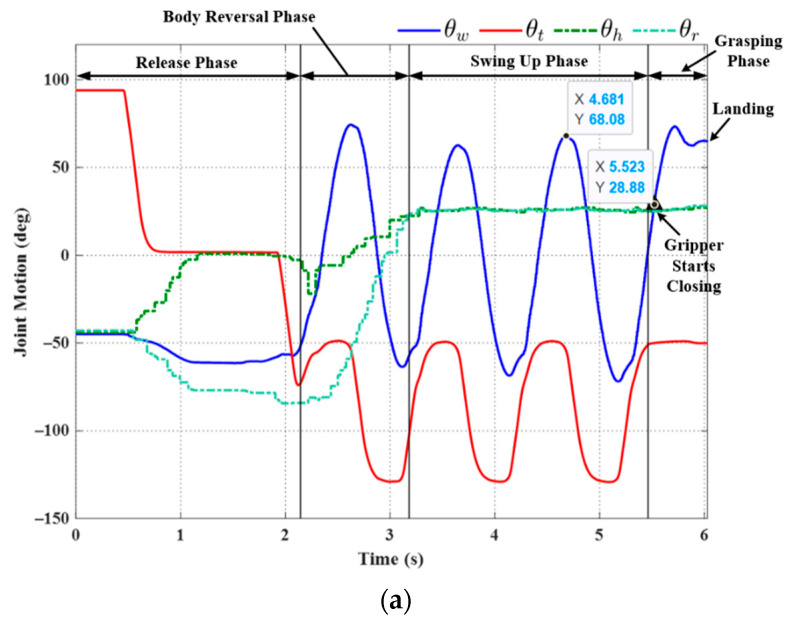
Experimental results (Exp. (g): gap distance: 250 mm, elevation: 0 mm): (**a**) joint motion; (**b**) predicted wrist joint motion and gripper command timing.

**Figure 24 sensors-22-04031-f024:**
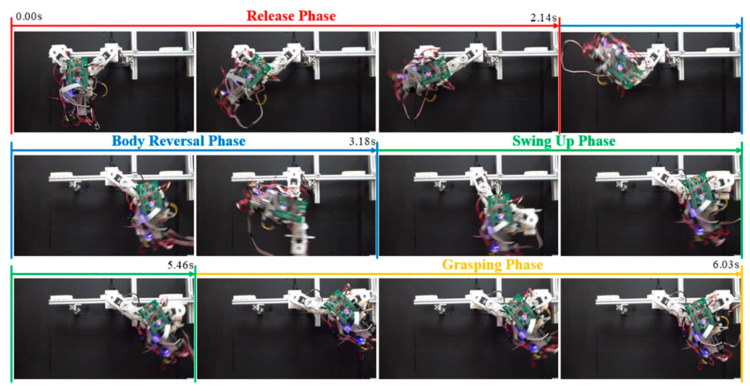
Snapshots showing the ledge brachiation experiment (Exp. (g): distance: 250 mm, elevation: 0 mm).

**Figure 25 sensors-22-04031-f025:**
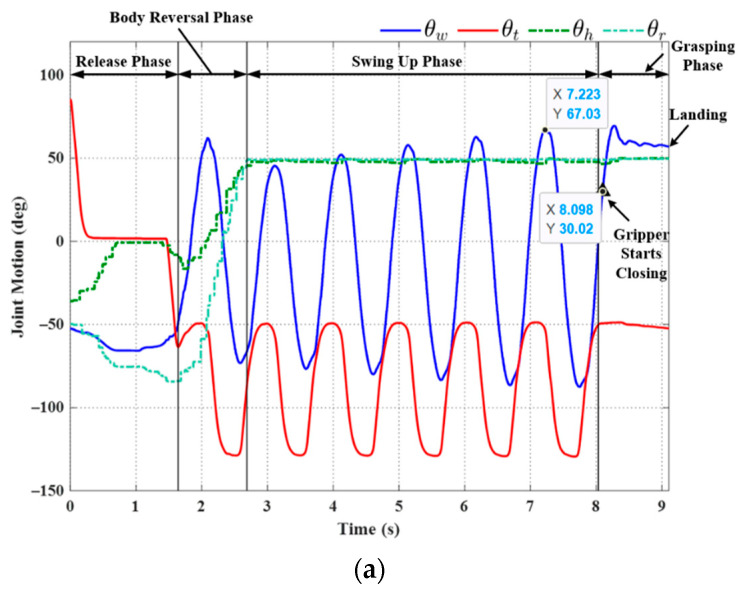
Experimental results (Exp. (h): distance: 192 mm, elevation: 56 mm): (**a**) joint motion; (**b**) predicted wrist joint motion and gripper command timing.

**Figure 26 sensors-22-04031-f026:**
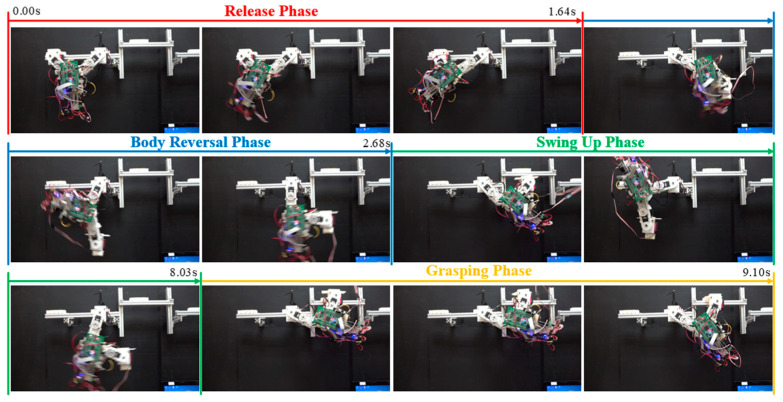
Snapshots showing ledge brachiation experiment (Exp. (h): gap distance: 192 mm, elevation: 56 mm).

**Table 1 sensors-22-04031-t001:** Comparison of brachiation robots and associated movement constraints.

	Proposed Robot	Gorilla Robot III [16]	MonkeyBot [23]	Tarzan [25]
Constraint:	Lack of joint actuation at gripper	Free rotation between gripper and grasped object	Flexible support	Free rotation between gripper and grasped object
Grasped object	Ledges on the wall	Horizontal bar	Liana vine (vertical rope)	Horizontal rope or cable
Orientation of grasped object	Parallel to the direction of movement and parallel to the ground	Perpendicular to the direction of movement and parallel to the ground	Perpendicular to the direction of movement and perpendicular to the ground	Parallel to the direction of movement and parallel to the ground
Anterior orientation relative to direction of robot movement	Transverse	Forward	Forward	Forward
# of actuations	3	3	1	1
# of links in modeling	4	4	2	2
Non-level horizontal brachiation	Yes	No	No	No
Susceptibility to gripper grasping dynamics and uncertainties	High	Low	Low	Low

**Table 2 sensors-22-04031-t002:** Parameters used in robot model.

Symbol	Physical Meaning
$m_{h}$	Mass of the support arm
$m_{b}$	Body mass
$m_{r}$	Mass of the release arm/gripper
$m_{t}$	Tail mass
$I_{h}$	Moment of inertia of the support arm
$I_{b}$	Moment of inertia of the body
$I_{r}$	Moment of inertia of the release arm/gripper
$I_{t}$	Moment of inertia of the tail
$l_{h}$	Length of the support arm
$l_{hc}$	Distance between the wrist joint and the center of mass of the support arm
$l_{b}$	Body length
$l_{bc}$	Distance between the joint of the release arm and the center of mass of the body
$l_{r}$	Length of the release arm
$l_{rc}$	Distance between the joint and the center of mass of the release arm
$l_{t}$	Tail length
$l_{tc}$	Distance between the body link and the center of mass of the tail
$\theta_{w}$	Angle between the support arm and the vertical line (wrist joint angle)
$\theta_{hb}$	Angle between the support arm and the body link
$\theta_{br}$	Angle between the body link and the release arm
$\theta_{bt}$	Angle between the body link and the tail link
$d_{w}$	Damping coefficient of the wrist joint
$d_{h}$	Damping coefficient of the support arm joint
$d_{r}$	Damping coefficient of the release arm joint
$d_{t}$	Damping coefficient of the tail joint
$\tau_{h}$	Torque given to the support arm joint
$\tau_{r}$	Torque given to the release arm joint
$\tau_{t}$	Torque given to the tail joint

**Table 3 sensors-22-04031-t003:** Parameter values of model used to describe robot dynamics.

Parameter	Value	Unit	Parameter	Value	Unit
$m_{h}$	0.217	kg	$l_{hc}$	0.055	m
$m_{b}$	0.141	kg	$l_{b}$	0.067	m
$m_{r}$	0.357	kg	$l_{bc}$	0.034	m
$m_{t}$	0.558	kg	$l_{r}$	0.102	m
$I_{h}$	$3.118\times 10^{-4}$	$\mathrm{kg}\cdot m^{2}$	$l_{rc}$	0.070	m
$I_{b}$	$1.051\times 10^{-4}$	$\mathrm{kg}\cdot m^{2}$	$l_{t}$	0.161	m
$I_{r}$	$6.462\times 10^{-4}$	$\mathrm{kg}\cdot m^{2}$	$l_{tc}$	0.067	m
$I_{t}$	$1.045\times 10^{-3}$	$\mathrm{kg}\cdot m^{2}$	$d_{w}$	0.011	N⋅m⋅s/rad
$l_{h}$	0.102	m			

**Table 4 sensors-22-04031-t004:** Experimental results obtained in the first set of experiments.

Experiment Set 1:	(a)	(b)	(c)	(d)	(e)	(f)
Success	No	Yes	No	Yes	Yes	Yes
Proposed gripper command strategy	No	No	No	Yes	Yes	Yes
Total time (s)	None	9.62	None	13.80	16.79	10.52
Max. swing angle (deg.)	46.55	48.05	46.45	60.35	57.46	56.16

**Table 5 sensors-22-04031-t005:** Parameter settings used in the second set of experiments.

Experiment Set 2:	(g)	(h)
Height difference	0 mm	56 mm
Gap distance	250 mm	192 mm
Desired arm joint angles ($\theta_{hb}^{f}$ and $\theta_{br}^{f}$) for grasping	26.22°	49.31°
Desired wrist joint angle $\theta_{w}^{f}$ for grasping	63.77°	56.95°
Compensation coefficient *a*	1	1.15
Max. swing angle during the swing-up phase	68.08°	67.03°
Final height	11 mm	59 mm
Transverse moved distance	247 mm	190 mm
Total execution time	6.03 s	9.10 s

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
