# Peer review of "Design of Transverse Brachiation Robot and Motion Control System for Locomotion between Ledges at Different Elevations"

_sensors, 2022, doi:10.3390/s22114031_

Round 1
Reviewer 1 Report
This paper designs a brachiation robot and proposes a motion control strategy that can switch in different stages. Finally, in the simulation and experiment, the robot both completed a cycle of moving along Ledges. In general, this paper presents interesting ideas and solution to resolve it. However, the contents need to be further improved, the following suggestions are available for the author's reference:
- Fig. 4(c) is an improvement on the structure shown in Fig. 4(b), which can effectively reduce the opening/closing power of the actuator. However, for the ledge with a column shape in Fig. 4(a), the structure shown in Fig. 4(c) is not applicable, which will cause deflection of the robot.
- When calculating the static constraint, the mass of the body link m2 is 0.141kg. However, it can be seen from Fig. 22 that when considering the devices installed on the body link, the mass is larger than 0.141kg, which will directly affect the checking of the maximum torque in Eq. (7).
- In the field of robot control, the dynamic expression of Eq. (11) is a very common form. And it is of reference significance only when the specific expression of four-link dynamics in Fig. 10 is derived.
- When calculating the torque generated by the tail link in Eq. (7), the position of the tail link's centroid should be at the midpoint of rod l4, not at the endpoint.
5 In this paper, the author mentions that "Note that the above experiment results are limited to one locomotion cycle, due to a lack of structure rigidity and the effects of cable winding when body flipping was implemented continuously ", this defect can be overcome by simulation.
- Some simulation parameter settings in Table 3 are different from the design parameters in section 2, especially mh.
- Line 293 on page 14, "Simplifying (1)" is incorrect.
- The "Introduction" of this paper needs further improvement. The number of papers approaching this topic is relatively small in literature, and the format of references is not uniform.
- There are some writing problems in this paper, for example, the full name should be given when the "COG" appears for the first time. Line 399 on page 18, "he motion control strategy used..." is incorrect, the author needs to refer to and correct.
Reviewer 2 Report
This work designed a ledge brachiation robot that can move transversely. The manuscript is well organized and easy to understand, especially the mechanical analysis, modeling and simulation are solid. However, some concerns should be addressed before publishing.
- The contribution of this work should be clearer. Authors claim that their ledge brachiation robot with an arm-body-tail configuration and a novel gripper design. However, the arm-body-tail configuration is existing in the previous works, while the discussion about the novel gripper design and corresponding advantages are missing.
- The author also claims that they present guidelines for the design and control of multi-locomotion brachiation robots. Is this specific to this work, or is there generality that can be extended to a class of brachiation robots? What is the difference between contribution 2 and contribution 3?
- Table 1 compares the typical brachiation robots and associated movement constraints, however, the discussion about this is missing, such as the advantages and disadvantages of each type. As mentioned before, the contribution and advantages of this work should be clearer.
- For the introduction section, authors are suggested to discuss the importance or potential application of brachiation robots, and the current challenging in this area.
- The locomotion control strategy of this brachiation robot is provided. Any controller or algorithm is adopted to improve the stability, accuracy or controllability of the system?
- Despite the brachiation robot seems successful from simulations and experiments, how to evaluate the experimental results standardly? When considering that there are some similar works, any quantitative comparison can be provided?
Reviewer 3 Report
The authors have developed a robot that moves by swinging between ledges and have proposed a motion control method. The effectiveness of the proposed motion has been verified by simulation and actual experiments. However, it is difficult to understand the novelty and usefulness of the research because the position of the proposed motion in relation to previous research is unclear.
Major comments
1-1 There are insufficient explanations to previous studies which are listed in Figure 1 and Table 1. In addition, the position of this study in comparison with previous studies is not clear. What is novelty of this study compared to other studies?
1-2 Can this method apply to a ledge with a step in the downward direction? If not, what are the possible problems?
Minor comments
2-1 The right edge of Figure 8(c) is missing.
2-2 The first “the” in Sec. 5.5 is a typo.
2-3 The first the at the beginning of Sec. 5.5 is a typographical error. (
2-4 The sentence is difficult to read because of many references to the equation behind the sentence (e.g., lines 382 and 394).
2-5 I did not understand the meaning of line 482. The maximum magnitude is not 32deg.
Round 2
Reviewer 1 Report
In this review round, the authors have well addressed all my concerns. I think this paper could be accepted. The quality of some figures in this paper should be improved. In particular, pay attention to the sharpness of the figures.